# Rehabilitation via HOMe-Based gaming exercise for the Upper limb post Stroke (RHOMBUS): a qualitative analysis of participants' experience

Cherry Kilbride [1], Tom Butcher [1,2] Alyson Warland,[1] Jennifer Ryan,[1,3] Daniel J M Scott,[1,4] Elizabeth Cassidy,[5] Dimitrios A Athanasiou,[4] Guillem Singla-Buxarrais,[6] Karen Baker,[1,4] Meriel Norris[1]

¹Department of Health Sciences, Brunel University London, Uxbridge, UK
²Department of Sport, Exercise and Rehabilitation, Northumbria University, Newcastle upon Tyne, UK
³Public Health and Epidemiology, Royal College of Surgeons in Ireland, Dublin, Ireland
⁴Neurofenix, London, UK
⁵Freelance Academic and Research Supervisor, London, UK
⁶Neurofenix, Atlanta, Georgia, USA

**Correspondence to**
Dr Cherry Kilbride;
Cherry.Kilbride@brunel.ac.uk

## ABSTRACT

**Objective** To report participants' experiences of trial processes and use of the Neurofenix platform for home-based rehabilitation following stroke. The platform, consisting of the NeuroBall device and Neurofenix app, is a non-immersive virtual reality tool to facilitate upper limb rehabilitation following stroke. The platform has recently been evaluated and demonstrated to be safe and effective through a non-randomised feasibility trial (RHOMBUS).

**Design** Qualitative approach using semistructured interviews. Interviews were audio recorded, transcribed verbatim and analysed using the framework method.

**Setting** Participants' homes, South-East England.

**Participants** Purposeful sample of 18 adults (≥18 years), minimum 12 weeks following stroke, not receiving upper limb rehabilitation prior to the RHOMBUS trial, scoring 9–25 on the Motricity Index (elbow and shoulder), with sufficient cognitive and communicative abilities to participate.

**Results** Five themes were developed which explored both trial processes and experiences of using the platform. Factors that influenced participant's decision to take part in the trial, their perceptions of support provided during the trial and communication with the research team were found to be important contextual factors effecting participants' overall experience. Specific themes around usability and comfort of the NeuroBall device, factors motivating persistence and perceived effectiveness of the intervention were highlighted as being central to the usability and acceptability of the platform.

**Conclusion** This study demonstrated the overall acceptability of the platform and identified areas for enhancement which have since been implemented by Neurofenix. The findings add to the developing literature on the interface between virtual reality systems and user experience.

**Trial registration number** ISRCTN60291412.

## INTRODUCTION

Despite advancements in prevention, treatment and rehabilitation, stroke remains a leading cause of disability worldwide[1] including in the UK where an estimated 77% of first-ever stroke survivors present

---

### STRENGTHS AND LIMITATIONS OF THIS STUDY

⇒ An in-depth qualitative study exploring participants' experiences of using a non-immersive virtual reality gaming platform for home-based upper limb rehabilitation following stroke.
⇒ A purposive sampling frame was used to capture the experiences of 18 participants with upper limb impairment ranging from mild to severe who had varying levels of engagement with the platform.
⇒ The framework method was applied to analyse the data through a rigorous iterative process of coding, revising and grouping by independent analysts overseen by a lead analyst.
⇒ Participants were interviewed by research team members that they may have interacted with during the trial.

---

with upper limb weakness.[2] Less than 20% of stroke survivors regain full function of the upper limb at 6 months.[3]

A combination of poor upper limb recovery and evidence that conventional rehabilitation results in insufficient upper limb training[4–6] has resulted in an increasing interest in alternative training programmes. Novel approaches include virtual reality (VR) platforms to intensify upper limb training,[7–9] while providing motivational feedback to encourage engagement and the required repetition to drive recovery.[10 11] Studies have demonstrated VR devices can encourage higher numbers of repetitions, provide immediate feedback on performance and stimulate the visual, auditory and tactile senses to increase neuroplasticity, therefore contributing to improvements in motor function and performance of daily activities.[12 13] Within qualitative work in the field a common theme that has emerged between studies is the beneficial effect VR has on motivation and engagement with upper limb rehabilitation.[14 15]

Initial evidence suggests such platforms are safe and effective at improving impairment, activity and participation, with the current Cochrane Review stating that VR may be beneficial for the upper limb when used as an adjunct to usual care; however, the evidence is currently considered to be of low quality.[7 16–20] A recent meta-analysis, which compared VR interventions that included gaming components with those which just provided visual feedback, found that the inclusion of gaming components produced larger treatment gains.[21]

Despite encouraging outcomes, platforms used to date are often inaccessible to stroke survivors due to cost and the physical demands of the user interface.[11 22–27] A recent systematic review of the acceptability of these platforms indicated several desirable features such as usability, small size, ease of set-up, sufficient support and engagement through variability, challenge and performance-based feedback.[28]

The Neurofenix platform (www.neurofenix.com) is a non-immersive VR therapy platform for gamification of poststroke upper limb rehabilitation using the Neuro-Ball, a novel hand-controlled gaming device. Developed by stroke survivors, physiotherapists and bioengineers, it delivers a safe, upper limb training programme, which has demonstrated positive effects on upper limb impairment and function.[29] However, for any platform to be integrated as part of standard care it must also be accessible and acceptable to the end users. This study aimed to explore the acceptability of using the VR platform for home-based upper limb rehabilitation within the context of a wider safety and feasibility trial with individuals in the chronic phase following stroke.[30]

## METHODS

This qualitative descriptive study was embedded in a non-randomised intervention design with a parallel process evaluation, exploring the participant experience and acceptability of using the VR platform, and what it was like to take part in the study. Qualitative descriptive studies are not based on a specific methodological approach and stay close to the surface of the data as described by participants.[31] Semistructured interviews were completed with a purposive sample of 18 participants from the intervention study. Participants were 39–85 years of age, were 1–7 years following stroke and represented various user engagement levels ascertained through data collected from the Neuro-Ball (high user defined as using ≥4 days/week). An interview guide was developed and piloted with service users; a copy of this is included in online supplemental material 1. Areas explored during the interviews were previous experience of upper limb rehabilitation, using the VR platform for home-based rehabilitation, perceptions about the supported remote training model used, barriers and facilitators to regular use and experience of trial processes. Interviews were conducted within 2 weeks of completion of the 7-week intervention at participants' homes. Carers or spouses involved were also invited to take part in a dyad if preferred by the participant, with all those who were interviewed providing written informed consent. A research therapist (TB or DJMS, both qualified physiotherapists) conducted the interviews. Continuity between interviewers was facilitated through shared piloting of the interview guide and initial review and agreement of the overall approach to the interviews. All interviews were audio recorded and professionally transcribed verbatim. The transcripts were anonymised prior to analysis.

### Patient and public involvement

Stroke survivors were involved in the iterative development of the VR platform. Two stroke survivors acted in an advisory capacity during the study, assisting with the development of the interview guide and providing input to the trial documentation and dissemination. Good practice guidelines available at the time for patient and public involvement were followed[32] and individuals were reimbursed for their time and expertise.

### Data analysis

Transcripts were analysed using the framework method.[33] The lead analyst (EC), an experienced qualitative researcher, read and re-read all transcripts noting initial thoughts and impressions of the data. Three transcripts were then read and independently coded by EC, applying a phrase or 'code' to important passages relevant to the research aims. An initial analytical framework based on the analysis of these three transcripts was then devised. The framework consisted of 50 codes grouped into 13 overarching categories. A brief description was written for each code to enhance the transparency of the coding process.

Three analysts (MN, DJMS, TB) then independently coded the same three transcripts and compared their codes to the initial framework. Minor adjustments were made to the framework because of this, resulting in the addition of three more codes. This second iteration of codes was then agreed on by EC, CK and MN and the working analytical framework was applied to a further three transcripts by EC. This resulted in further minor changes to the working analytical framework. This coding framework was agreed by EC, CK and MN and applied to the remaining 12 transcripts by EC. The iterative process of coding, revising and grouping continued until no new codes or categories were generated. The final framework consisted of 60 codes.

The data were then summarised in a matrix using Microsoft Excel. Narrative summaries were composed and exemplar quotations were identified for all categories. Subthemes and themes were then inferred from the matrix by reviewing the data and connecting related ideas and concepts.

## RESULTS

Participant characteristics are summarised in table 1. The average interview time was 63 min (range 28–88 min).

**Table 1** Interview participant characteristics

| Participant number and pseudonym | Side of hemiplegia | Simplified modified Rankin Score* | Fugl-Meyer Assessment Upper Extremity (FMA-UE)† | High/low‡ user of NeuroBall |
|---|---|---|---|---|
| 1. Iris | Right | 2 | 18 | Low |
| 2. Lina | Right | 2 | 32 | High |
| 3. Ray | Left | 3 | 57 | High |
| 5. Pam | Right | 2 | 29 | High |
| 6. Mark | Right | 1 | 53 | Low |
| 13. Steve | Right | 3 | 16 | High |
| 16. Ann | Left | 3 | 31 | High |
| 17. Tina | Right | 2 | 63 | Low |
| 18. William | Left | 3 | 48 | High |
| 19. Sam | Left | 3 | 35 | High |
| 20. John | Left | 2 | 58 | High |
| 22. Bal | Left | 3 | 33 | High |
| 23. Sue | Left | 5 | 8 | Low |
| 24. Linda | Right | 3 | 36 | Low |
| 25. Ed | Right | 2 | 40 | High |
| 26. Mike | Right | 2 | 12 | High |
| 27. Elaine | Left | 2 | 15 | Low |
| 28. Terry | Right | 3 | 8 | High |

*Score of 2=slight disability, 3=moderate disability, 4=moderately severe disability, 5=severe disability.[40]
†Score of 0–28=severe impairment, 29–42=moderate impairment, 43–66=mild impairment.[41]
‡High user=4 or more times a week, low user=3 or less times a week.

## Summary of themes

Five themes were developed from the data, summarised in table 2. Findings offer an insight into the range of participants' views and a sense of convergence and divergence. Exemplar participant quotes are provided with details of their Fugl-Meyer Assessment Upper Extremity (FMA-UE) score, amount of device use to add context and transcript page and line numbers.

### Theme 1. Trial enrolment: influencing factors

All participants regardless of the severity of their upper limb impairment enrolled in the trial hoping to improve their upper limb recovery. Most described under-resourced and inadequate rehabilitation services for the upper limb following stroke. In their experience, time-limited rehabilitation efforts had been directed at achieving functional gait at the expense of upper limb training and recovery.

> P24 Linda (FMA-UE 36, low user): [acute setting] I was just sitting there looking at the walls (33, 1515). […] I had plenty of time because I was, I was there, sitting there, thinking, 'what can I do to move? What can I do to move my hands and that?', and there was very little help, very little help (34, 1544–6).

Positive reports of acute rehabilitation were scarce. Likewise, community therapy was described as usually time limited and mainly focused on functional gait. Just one participant described having publicly funded upper limb rehabilitation in the community. A sense of disappointment about the lack of poststroke rehabilitation services offered in acute and community settings pervaded these accounts.

Many of these stroke survivors had adapted or created their own exercise programmes and sought out alternative interventions and services, such as electrical stimulation, to fill perceived gaps in service provision. Where described, these exercises appeared to be non-specific, low intensity, self-devised and poorly structured.

> P5 Pam (FMA-UE 29, high user): …I don't do much… umm, yeah, the turning, having my fingers out straight and bending back […] and trying to straighten out my arm, move it round, okay, umm, yeah, and things like that (9, 350–61).

Home exercise programmes were hard to sustain for all but the most determined participants without ongoing feedback and encouragement.

### Theme 2. Perceptions of pretrial preparation, in-trial support and communication

Participants reported enough information was provided about the study before enrolment. All participants agreed the training about the games and using the device was thorough and delivered at the right level for their individual technical ability and experience. The amount and

**Table 2** Themes and subthemes

| Themes | Subthemes |
|---|---|
| 1. Trial enrolment—influencing factors | 1.1 Poststroke rehabilitation<br>1.2 Aspirations for upper limb recovery<br>1.3 Failure of self-devised home training programmes<br>1.4 Previous experience with trial devices and technology |
| 2. Perceptions of pretrial preparation, in-trial support and communication | 2.1 Pretraining advice, support and information given<br>2.2 In-training support and communication<br>2.3 Involvement of family members |
| 3. Device usability and comfort | 3.1 Set-up and fitting the NeuroBall device<br>3.2 Equipment failure |
| 4. Factors motivating persistence | 4.1 Utility<br>4.2 Game preferences<br>4.3 Incentives, feedback and family support |
| 5. Perceived effectiveness of the intervention | 5.1 Improvement in upper limb activity<br>5.2 Reasons for perceived effectiveness of the intervention<br>5.3 Reasons for perceived lack of effectiveness of the intervention<br>5.4 Impact of participation in the trial on post-trial exercise |

type of training was varied for individual needs. At the end of the training, participants reported having enough confidence to start playing the games.

> P16 Ann (FMA-UE 31, high user): [the research assistant] went through everything. He went through all the games as well, which was good, so that helped. So, it just gives you confidence and reassurance in what you're doing (7, 266–8). They were all pretty, you know, self-explanatory in the end but it was good to go through it (7, 274–8).

Participants and /or their carers were satisfied with the level of support offered during the trial and the ease at which they could contact the research team if help was needed. The list of contact details offered reassurance that help was readily available.

Eight participants either needed no help with the device (P1, P2, P17, P22) or just advice or support from a family member for example, to charge the device (P16), or to call the research team for help (P5, P6, P16, P24). In this group of eight, upper limb impairment ranged from mild to moderately severe (FMA-UE 18–63) as defined by Hoonhorst *et al*.[34] Seven participants needed physical assistance from family members to set up the device, put the device on their hand, or secure the straps.

> P19 Sam (FMA-UE 35, high user): One of my [family members] told me [how to put my hand in]. So, I was alright then (11, 471). […] definitely useful to have

somebody [family member] around in case you don't use it right (36, 1684–5).

Both participants with severe impairment (P23 FMA-UE 8, low user; P28 FMA-UE 8, high user) reported needing the most physical assistance which may have contributed to P23 Sue being a low user; however, it was not a barrier to engagement for P28 Terry who was a high user. While people with a severe impairment may need help using the device, as indicated in our findings some people with moderate or mild impairment may also need assistance at least when they first start to use the device.

### Theme 3. Device usability and comfort

All participants but one (P23 FMA-UE 8, low user) were able to use the NeuroBall device with relative ease. Nevertheless, some issues were raised. Difficulty getting the fingers or thumb into the device due to spasms or stiffness hampered the initial set-up for some participants.

> P1 Iris (FMA-UE 18, low user): Well it's hard to get the [fingers in] … 'Cos first of all you have to try and push it down … And then you get some spasms in your fingers, so you have to prise your fingers out (13, 544–7).

Iris reported it could take up to 8 minutes to get the hand into the device. For some participants fitting improved with practice (P5, P19, P20, P22, P24); for others, it remained a tricky and time-consuming process throughout the trial (P1, P2, P6, P16).

The straps and hooks, which secure the hand to the NeuroBall and the NeuroBall to its base to aid calibration, were described as a bit fiddly, and tricky to use with only one hand (P1, P2, P3, P5, P6, P16, P17, P19, P22, P23, P25, P26, P27, P28). However, participants described getting better at this with practice or simply avoiding using the hooks.

The majority of participants (n=14) experienced varying issues with the equipment, most of which were related to the app which were resolved through a basic restart. Occasionally, problems with the NeuroBall device itself arose.

> P3 Ray (FMA-UE 57, high user): […] the first couple of weeks or three weeks very good, but then of course towards the end, the last couple of weeks, umm, that middle finger, something broke inside. I hadn't dropped the ball in any way […] obviously there's a weak link in there (14, 617–23).

Most technical problems were resolved with advice from the research therapists or engineers if needed. The most common problem reported with the NeuroBall was damaged, broken or ineffective straps which secure the hand to the device; however, resolutions were always found to allow participants to continue with their training.

### Theme 4. Factors motivating persistence

Participants used the VR platform for a median of 17.4 hours over 7 weeks.[29] Several reasons were highlighted that encouraged this engagement.

The majority of participants (n=12) found the computer tablet's touch screen easy to navigate.

P26 Mike (FMA-UE 12, high user): Yeah, got the handle of it pretty quickly […] I was able to, to control it. […] Yeah, it was easy enough to, to, to select the games I needed to, yeah, yeah (44, 2044–52).

While three participants struggled initially, difficulties with navigation were quickly overcome and were not related to previous touch screen experience.

Most participants liked the therapeutic games and enjoyed playing them. The most popular games were Scuba Diving (n=9), Holidays Jogger (n=9), Space Shooter (n=8) and Pong Goal (n=7). Preferred games either encouraged specific movements or activated specific actions, were associated with previous hobbies or interests, offered the right level of challenge or were fun and absorbing. An appropriate level of challenge was somewhat limited for those with mild impairments and resulted in limited motivation and below-average gameplay. This was not seen in those with moderate to severe impairment, despite some reporting that they experienced monotony. A summary of all game preferences (likes and dislikes) is reported in online supplemental material 2.

Most participants found the incentives, rewards, encouragement and feedback built into the VR platform and the games to be positive features. Reassuring messages encouraged persistence or motivated participants to improve their score or to repeat a good performance.

P24 Linda (FMA-UE 36, low user): [Ten in a row! Well done!] Oh that was encouraging, that was encouraging it all […] it was very encouraging […] It was making want to do more and more […] because it was encouraging and it was speeding me on to do it (26, 1185–97).

Likewise, participants commented on the inherent competition created by the leaderboard and how that spurred them on to achieve more. Other objective feedback (eg, game difficulty level, number of repetitions and minutes played) was also well received and used as a target or benchmark for current or future performance and effort.

P19 Sam (FMA-UE 35, high user): I, umm, enjoyed playing on it and I always attempted to go a little bit longer. If I'd done thirty minutes, next day thirty-five minutes - next day forty minutes (27, 1201–3).

Training that focused on hand movement and action repetition were also cited as factors in persistence. Some motivators related to enjoyment, such as playing games connected to previous interests (eg, football, playing space invaders as a child) and finding gaming more interesting and purposeful than prescribed home exercise programmes. Other motivators were logistical, such as having a structured practice schedule and set amount of time to practise, being able to play at home and having the flexibility to plan practice around daily life.

P22 Bal (FMA-UE 33, high user): I liked the fact that, because it was a set period and, er, I was motivated to do it every day, er, and I set aside time to do the exercises regularly […] The fact that it was at home, I could plan my day around… all my exercises around my other activities. […] So that was very useful […] the fact we can do it at home (16, 690–704).

Several factors were identified which could have further increased motivation to use the platform such as a wider range of games and greater control over the level of difficulty.

P22 Bal (FMA-UE 33, high user): If there was a bit more variety or slightly different games, or even the games you could adjust so that, er, it was, er, slightly different, then I think it might not be so boring (15, 656–8). […] because of the lack of variety, er, I found them a bit boring after initial excitement (17, 738–9).

### Theme 5. Perceived effectiveness of the intervention

The majority of participants reported some beneficial effects such as a perceived reduction in stiffness, lower odds of having shoulder pain and improved range of movement in the shoulder, elbow and wrist.[29]

P27 Elaine (FMA-UE 15, low user): Well, it…it just feels more relaxed and…it doesn't…it doesn't hurt so much. I feel more…like now I am using…actually using my arm to do things, more than my…just swinging my shoulder round (14, 627–9). I think it's improved the strength in that left arm (16, 708).

Improved function for tasks such as dressing and washing up was reported by two participants. For others, playing the games prompted them to try to use the upper limb more in everyday life.

P20 John (FMA-UE 58, high user): I think what it might have done was make me think more carefully about things where I can use my left hand, I thought to myself, 'I'm not using this left hand enough.' So, when I can I tend to use it (18, 779–81).

This positive linking between practice and functional use may have increased psychological investment in the potential of the game to have positive outcomes, which in turn may have increased persistence in training. Playing enjoyable and immersive games focused attention on the game rather than the purpose of the game (ie, repetitive upper limb exercise). When compared with conventional exercise programmes, captivating, interesting and enjoyable games have sustained longer training periods.[7] Seven participants reported gaming had a positive effect

on their motivation to do their previous home exercise programmes or made the exercises more manageable.

> P13 Rose (Steve's (FMA-UE 16, high user) wife): […] he seemed to be a lot more motivated to do the […] exercises after the NeuroBall, I think. […] 'Cos you've sort of got into the routine of doing quite a lot so then you sort of have carried that on. So yeah, it's quite good (3, 81–92).

Poor outcomes were attributed to the trial not being long enough for people with more severe impairment who believed a longer trial was needed to deliver positive outcomes.

> P26 Mike (FMA-UE 12, high user): I'll go back to the length of time, the time was, was kind of limited, but my arm and the, the, the damage done to it by the stroke, umm, it seems just something that needs a more prolonged piece of work (46, 2135–7). I needed more prolonged work. If… I might have seen a lot, a lot more difference (46, 2146).

In-game difficulty levels were not perceived to be high enough to drive improvements for those with mild impairment. Participants also suggested the complex movement of the wrist and hand could not be retrained sufficiently by the limited number of movement options offered by the device.

> P19 Sam (FMA-UE 35, high user): I don't think it twists your hand enough (16, 719). I get to the limit and then the machine doesn't make it any better (17, 724–5).

## DISCUSSION

These findings illustrate that the VR platform was acceptable to the participants and addressed several requirements for useful technology-facilitated upper limb rehabilitation suggested by previous authors.[28] While some participants raised issues related to the comfort and ease of use of the device, these mostly did not appear to impact the use of the platform or enjoyment of the games. Getting their hands into the device was the main issue raised; however, participants reported this got better with small adjustments and practice. Most technical problems were resolved with the help of the support team including the research therapists and engineers.

Practically, participants reiterated the importance of the adaptability of the VR platform to the home environment, both in terms of space and flexibility to use it around the demands of everyday life. The importance of this integration is noted by others including Wingham et al[24] who identified five themes affected by this experience: diligence of play, perceived effectiveness, acceptability, caregiver and social support, and the set-up and administration of the rehabilitation platform. These experiences and perceptions of the participants created a more engaging and flexible rehabilitation environment

and resulted in more frequent usage of the rehabilitation platform in a home environment. However, Standen et al[35] reported competing commitments as a barrier to usage at home even though the rehabilitation platform was perceived as flexible and motivating by participants.

The VR platform was perceived in most cases to deliver sufficient motivation to encourage perseverance. Visual feedback of movement success and motivational factors such as objective measures were important to derive a sense of achievement and the drive to continue. Another factor that appeared as being important in persistent use was the belief that the intervention had the capacity to help, creating psychological investment in the platform.[36] Interest was also raised through a sense of challenge. Such findings reiterate the importance of features to enhance engagement noted by others.[11 22 24 27 28 35] An expanded variety of gameplay, specifically the level of challenge (including more complex movements), motivational tools such as a real-time leaderboard and the addition of more complex hand movements were identified as factors to improve engagement in future iterations of the device. These findings highlight the critical importance of user engagement in technology development and ensuring all levels of ability are included. Some suggestions do however need to be considered with caution. The concept of a real-time leaderboard may promote social connectivity through competition and is aligned with the importance of social interaction in gameplay.[28] However, it could potentially lead to disengagement for those who fail to win.[11 24 27] Opt-in competitions with others may be more appropriate than compulsory competitive features. Multiple, personal, interacting factors such as preference of game genre, level of difficulty and movements required for the game are likely to contribute to extended gameplay and persistence, therefore flexibility in the platform to adapt to preferences would be an important development.

Perceived improvements in movement, function and awareness of the upper limb increased gameplay motivation and use. This concurs with observations from Standen et al[35] and Wingham et al[24] and relates to previous rehabilitation experience of the participants. Most had been dissatisfied with previous therapy and lack of progress in upper limb function. As a consequence, it is perhaps unsurprising that hope for further recovery would manifest in motivation to persevere when positive changes were perceived.

Theme 2 highlighted the importance of access to support and confidence in troubleshooting. In agreement with reported findings,[24 28 35 37] our findings reiterate the importance of ongoing support for day-to-day use as well as within a trial. Available support should be considered, particularly for those with more severe upper limb impairment. Participants who had difficulty fitting the device may have benefited from further training on stretching and managing spasticity and spasms. Participants appreciated the clear signposting for support provided by the research team and in the handbook, which is important

given other authors have suggested that accessing of support may require proactive encouragement.[35] Interestingly, the need for additional support in this study was not associated with previous use of similar technology. While this conflicts with other studies,[28 35] it may indicate the importance of clear and adequate training and access to clear instructions for all.

## Strengths and limitations

Participants were a representative sample of those who trialled the intervention, covering a range of ages, severity of upper limb impairment, usage and gender. Descriptions of disappointing community stroke rehabilitation are consistent with previous accounts.[38] Their interest in pursuing upper limb recovery in the chronic stages following stroke is also not unusual.[39] Participants were characteristic of many stroke survivors living in the community, and their experiences of taking part in the trial provide useful insights about the acceptability and utility of home-based VR training and can inform future trial design.

Participants were interviewed by research team members that they may have interacted with during the trial. These interactions may have impacted their responses to specific topics both positively and/or negatively. Nevertheless, a wide range of views were evident through the interviews which would indicate a willingness to share experiences.

A rigorous and transparent analysis was conducted with a clear audit trail. However, no participant validation was conducted which could have enhanced insights further.

## CONCLUSION

This study illustrates the complex interactions that users have with tools such as the Neurofenix platform. Findings clearly demonstrated the platform's acceptability and identified positive design and functional features, highlighting the need for adaptability to individual requirements and preferences. Feedback received has already resulted in significant developments to the VR platform including to the physical fit of the NeuroBall device, the platform usability and the development of further training and support materials. Participants highlighted the importance of meaningful motivation to enhance engagement through features such as challenge, competition, provision of fun activities and feedback which can indicate palpable positive change. Overall findings indicated that the VR platform delivered on most of these features and provided clear indications for future development and add to the developing literature on the interface between VR and the user experience.

**Acknowledgements** The authors thank all participants in the RHOMBUS trial, the two stroke survivors who assisted the development of the interview topic guide and advised on the trial documentation and dissemination. Further thanks to the group facilitators of Different Strokes and the Action for Rehabilitation from Neurological Injury.

**Contributors** CK and GS-B conceived the study. CK, TB, AW, JR, DJMS and MN designed the study. GS-B, DAA, DJMS, TB and CK designed the intervention. DJMS and TB collected the data. EC led the data analysis supported by MN, TB, DJMS and CK. MN and TB prepared the manuscript. MN, CK, TB and KB contributed to drafts of the paper and approved the final draft. All authors read and approved the final manuscript. CK is the guarantor for this study.

**Funding** This work was supported by Innovate UK (grant number: 104188).

**Competing interests** GS-B and DAA are employed by Neurofenix, a digital therapeutics company developing a therapy platform and sensor-based devices to augment rehabilitation. Neurofenix had no influence on the design of the study, data collection, analysis and interpretation of the data or manuscript preparation. KB and DJMS were employed by Brunel University London during the period of the study and are now Neurofenix employees.

**Patient and public involvement** Patients and/or the public were involved in the design, or conduct, or reporting, or dissemination plans of this research. Refer to the Methods section for further details.

**Patient consent for publication** Not applicable.

**Ethics approval** This study involves human participants and was approved by the College of Health and Life Sciences Research Ethics Committee (REC) in Brunel University London (10249-MHR-Mar/2018-12322-2). Participants gave informed consent to participate in the study before taking part.

**Provenance and peer review** Not commissioned; externally peer reviewed.

**Data availability statement** No data are available. Participants did not consent for datasets to be stored or accessed outside of the research team. Therefore, no datasets have been made publicly available.

**ORCID iDs**
Cherry Kilbride http://orcid.org/0000-0002-2045-1883
Tom Butcher http://orcid.org/0009-0005-4452-4679

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
