## [Reviewer comments · BMJ Open]

ARTICLE DETAILS

TITLE (PROVISIONAL)	Rehabilitation via HOME Based gaming exercise for the Upper-limb post Stroke (RHOMBUS): a qualitative analysis of participants' experience.
AUTHORS	Kilbride, Cherry; Butcher, Tom; Warland, Alyson; Ryan, Jennifer; Scott, Daniel J M; Cassidy, Elizabeth; Athanasiou, Dimitrios; Singla-Buxarra, Guillem; Baker, Karen; Norris, Meriel

VERSION 1 – REVIEW

REVIEWER	wang, rui West China School of Nursing, Sichuan University/Department of Neurology, West China Hospital, Sichuan University
REVIEW RETURNED	30-Jun-2023

GENERAL COMMENTS	Virtual technology can improve the accessibility of post stroke rehabilitation and greatly assist patients. But I must say that I am sorry that I do not have experience in writing qualitative research. From my perspective, this paper lacks clarity in explaining qualitative research methods, and there are also certain limitations in selecting and refining research topics. It is recommended to further improve it.
--

REVIEWER	Amorim, Paula University of Beira Interior, Health Sciences Faculty
REVIEW RETURNED	06-Jul-2023

GENERAL COMMENTS	.Title «HOME» seems to have more capital letters than it should (Home instead of HOME). The title doesn't mention the phase after stroke (Chronic phase). 2. Introduction It misses a theoretic basis for Acceptability (eg, theory of planned behaviour, technology acceptance model (TAM), unified theory of acceptance and use of technology [UTAUT]). Additional external factors influence the user acceptance of technology, including demographic characteristics, benefits experienced with technology, technological expertise, and social/cultural influences. Usability, accessibility, comfort, privacy and security, satisfaction, are also important variables to the acceptability of telehealth solutions. The introduction should go into more depth. 3. Methods I don't agree with the sentence «qualitative descriptive studies...stay close to the surface of the data».
---

	Qualitative research aims to obtain a deeper and more holistic understanding what health professionals, patients or stakeholders think, feel or do in their natural context, through collection and analysis of narrative material, using a flexible study design. Methodology in qualitative research differs in many aspects from quantitative research. In qualitative studies, the questions are broader and open to unexpected discoveries. This flexibility is a strength of qualitative research but only within a coherent methodology as a whole. Qualitative research prefers the question SPIDER (Sample, Phenomenon of Interest, Design, Evaluation, Research Type) over PICO (Population, Intervention, Comparison, Outcome) Main differences between quantitative research and qualitative research   Quantitative research Qualitative research   Investigation question: PICO Quantitative research asks «how much», «how many», «how often», The investigation question remains the same through the research process Investigation question: SPIDER Qualitative research asks «what», «how», and «why» During the research process the research question might change to a certain degree because data collection and analysis sharpens the researcher's lens. In the methods section you need to describe how and explain why the original research question was changed. Sample and selection: The sample size is larger and calculated with statistical rules in a way to get a representative sample of the population. The ideal is alleatoy recruitment. Once the sample size is determined, it is intended to maintain this size until the end od the study; if that is not possible (due to drop-offs, serious secondary effects, ...) in method section you must explain the reasons. Sample and selection: Small samples and not randomization in the seletction process. The choise of participantes should be made by individuals or groups who have knowledge on the phenomenon and can articulate and reflect, are motivated to communicate at length and in depth with the investigator and represent the stakeholders. The sample could change during the research process and the size is only closed when information saturation is reached (when it is expected that adding new participnats does not add new information but yield redudant information). You should review the sampling plan regulary and adapt when necessary. Comparison between intervention group and control group   There is no control group Observation in natural context   Manipulation of the context occurs Focus on quantification of predefined variables Open to new findings The aim is generalizaion of the resuts to the population Generalization is not intended. The aim is to achive a deeper understanding of the phenomena There is an effort to avoid researchers' influence (blinding effect) Researchers' influence is inevitable Sampling: the authors have chosen a convenience sampling, selecting the participants that are most easily accessible.	Quantitative research	Qualitative research	There is no control group	Observation in natural context
Quantitative research	Qualitative research				
There is no control group	Observation in natural context				

	Qualitative data collection method: the authors have chosen the interviews as the collection method. Analysis: The authors developed an inductive coding scheme, a process of open coding, creating categories and abstraction by formulating a general description of the phenomenon under study: subcategories with similar events and information are grouped together as categories and categories are grouped as main categories. The sampling saturation is reached when no further sampling is necessary (this should be mentioned by the authors). The authors indicated that they used an audit trail as a technique to enhance trustworthiness; they don't say if they used researcher triangulation (all data was analysed involving several authors) which is another technique to enhance trustworthiness. 4. Results A narrative research approach was followed to describe the results. Table 1 shows that all the participants had their stroke event more than a year ago, at least. This is the first time that the reader is aware of the context of the study: chronic phase after stroke. This is important information that should be in the title, in the methods and in the discussion. 5. Discussion The discussion should go into more depth. The authors don't integrate the results in the theoretic frameworks for Technology Acceptability. The authors don't say anything about the influence of time since stroke. Patients with 1 year since stroke feel the same way than patients with 7 years since their stroke?
--	---

REVIEWER	Dorsch, Simone South Western Sydney Local Health District, Bankstown-Lidcombe Hospital
REVIEW RETURNED	25-Jul-2023

GENERAL COMMENTS	Abstract Clearly written except Results section - particularly lines 64-66 could be re-written to be clearer, maybe specify or provide some example of the Influencing factors, maybe commence sentence with 'Participants' overall experience as influenced by.....' and the following sentence with 'Usability and acceptability of the platform was affected by.....' Introduction Clearly written overall. Please include the research aims/questions in further detail Suggestions are: Lines 97-98: specify if this is full function of the arm Lines 105,106 – some terms unclear – what is meant by motor feedback and multisensorial stimulation Line 107 – should read improvements in performance of daily activities Line 109 – larger than what? Line 120 – contains the terms 'upper-limb' and 'arm' - use consistent terms - I prefer 'arm' because it is more accessible language
---

	Methods Clearly written Comments and suggestions: I think that the reference 29 describes the method as being 'qualitative descriptive study', rather than the absence of a framework, this statement about not having a methodological framework appears to be contradicted in the Data Analysis section where the use of the Framework method is described Lines 133-134 – more detail about how many participants across what levels of engagement should be provided Include ethics clearance details Line 152 – these research aims have not been fully described Results Clearly presented Suggestions: Line 286 – improve expression of this sentence Line 290 – delete 'as illustrated previously' Line 301 – delete 'as stated' Line 307 – why is it 'perceived monotony' - this is describing a subjective experience and could be written as 'some commenting on monotony' Line 351 – unclear Lines 363-364 – is this referring to other research? Needs referencing Lines 382-385 – it would be good to have more detail about these important comments and supporting quotes, if identification of barriers is a research aim then these comments are very important Discussion Clearly written - however some of the concepts discussed here are not evident in the Results Suggestions: Line 391 – statement appears contradictory - does it mean that these issues only affected a small number of people or that they did not affect the amount of use? Line 411-412 – needs re-writing to increase clarity Lines 415-417 – these interesting factors are not well described in the Results section Lines 423-424 – what are these 'multiple, personal interacting factors' Line 433 – this is not apparent in the Results section Lines 434-435 – there is a very large body of evidence on neuroplasticity that describes these features of changing function after stroke Lines 445-446 – unclear Line 456 – could be the 'participants were representative of stroke survivors living in the community ...' Line 457 – the argument appears to be that the results should be fairly generalisable to similar groups of stroke survivors Conclusion Line 474 - typo Lines 476-478 – unclear writing Lines 478-480 – needs re-writing – what 'features' are being described? Acknowledgements - should the stroke survivors involvement with the study design be acknowledged with co-authorship?
--	---

REVIEWER	Bentley, Paul Imperial College London
REVIEW RETURNED	03-Aug-2023

GENERAL COMMENTS	Overall comments: While this work is interesting, I do not feel it is novel, nor does it add new findings or recommendations to the field. The presentation of this work needs to be improved to reach a more appropriate standard for peer-reviewed publication, in terms of formatting and grammar, as well as the rigour of scientific reporting standards. In particular, detail is lacking regarding the qualitative methods and analysis. Major comments:  • Please follow CONSORT reporting guidelines (state that this is a feasibility trial) in the title and throughout the manuscript. • The Background does not offer a balanced review and critique of relevant literature, please comment on the quality of evidence presented in systematic reviews and meta-analysis included in your manuscript. It may be worth including some key preclinical evidence and high-quality evidence synthesis (including Cochrane reviews). • Considering this is a quantitative study, there is limited to no discussion of existing qualitative work in the Background. The Background could be improved by exploring the wealth of existing qualitative work in this field and discussing how the present study builds upon this. • Research aims are not clearly stated within the manuscript. Aims presented within the Abstract are too vague ("To report participants' experiences of trial processes and use of the Neurofenix 49 platform for home-based rehabilitation following stroke") and not reflective of the methodology employed. • Methods section- please ensure you follow qualitative reporting standards i.e COREQ checklist. Key information is missing, such as sampling procedures (for both recruitment to the main trial and the purposive sampling method). Please state your theoretical assumptions- ontology, epistemology. • Methods- no reference to underlying theory used to inform interview guide development or framework development. Please clearly explain what guided the development of these during interactions with PPI representatives- i.e what were some of the key starting blocks. • Analysis- no reference to themes within the analysis, mentions "12 categories", Results section introduces "5 themes". Please discuss how these themes were generated in the analysis section, make clear the distinction between "categories" and "themes". • Discussion- not in-keeping with study aims or results (limited reference to trial processes). Would benefit from further review of relevant background literature and theory to demonstrate where the current findings sit and what they add.
---

	 • Conclusion lacks clarity, and again, is not in-keeping with the main study aims. There is limited critical insight from the authors in terms of limitations of this work. There needs to be more generalisable conclusions/implications drawn for this work to hold relevance to a broad readership. • Reference to the technology brand (Neurofenix) is featured throughout this manuscript, as well as a link to the company website. I would suggest referring to the brand name in the first instance only, and thereafter, referring to it as a VR platform. Authors competing interests make this particularly pertinent. • Please include coded transcripts, as well as map of codes, sub-themes and themes. • Minor comments:  • Please include a background in the Abstract to provide context for readers • Please label both strengths and limitations and ensure both are equally represented. The first strength listed is a description of the study rather than a strength, please elaborate or clarify this point. • Please ensure key aims, methods, results and conclusions are succinctly and cohesively presented within the abstract. • Grammar and presentation need attention to ensure clarity of expression and enhance readability. • Pg4, line 10 “Despite advancements in prevention, treatment and rehabilitation”, please reference • Pg4, line 42, it is not misleading to state that the platform in question is effective in generating improvements in at the level of impairment and function based on a small, non-randomised feasibility study. Please amend. • Pg5, line 51; Please state PPI guidelines followed, including reimbursement.
--	--

VERSION 1 – AUTHOR RESPONSE

Reviewer: 1

Mrs. rui wang, West China School of Nursing, Sichuan University/Department of Neurology, West China Hospital, Sichuan University

Comments to the Author:

Virtual technology can improve the accessibility of post stroke rehabilitation and greatly assist patients. But I must say that I am sorry that I do not have experience in writing qualitative research. From my perspective, this paper lacks clarity in explaining qualitative research methods, and there are also certain limitations in selecting and refining research topics. It is recommended to further improve it.

Thank you for your review and your comments. Unfortunately, as we are restricted on the available

word count to provide a more detailed explanation of the qualitative methods, we hope that the amendments that have been made as part of the overall review will have improved the clarity.

Reviewer: 2

Dr. Paula Amorim, University of Beira Interior

Comments to the Author:

Please read the attached file

1. Title «HOME» seems to have more capital letters than it should (Home instead of HOME).

Thank you for your review and comments. The capitalisation of O and M in HOME is part of the study name/anagram of RHOMBUS.

The title doesn't mention the phase after stroke (Chronic phase).

Thank you and we agree with your point regarding the title not mentioning the chronic phase after stroke. However, as this title aligns with the already published main paper and protocol for continuity purposes, we will not be able to change it. However, we have taken on board your point and will address this in the RHOMBUS II follow up trial and will ensure sub-acute stroke is in the title.

2. Introduction It misses a theoretic basis for Acceptability (eg, theory of planned behaviour, technology acceptance model (TAM), unified theory of acceptance and use of technology [UTAUT]). Additional external factors influence the user acceptance of technology, including demographic characteristics, benefits experienced with technology, technological expertise, and social/cultural influences. Usability, accessibility, comfort, privacy and security, satisfaction, are also important variables to the acceptability of telehealth solutions. The introduction should go into more depth.

Thank you and yes, we agree with this point made and in our follow-on study (RHOMBUS II) we have used the Theoretical Framework of Acceptability (Sekhon, 2017). However, at the time when we were designing this study (2016/17) there was no such agreed definition or framework to follow. We did use the QUEBEC measure of acceptability as a quantitative measure of acceptability. As can be seen from the included interview topic guide many of the 7 themes outlines in the Theoretical Framework of Acceptability were covered, just not explicitly by design. Some of these details are also covered in the published quantitative paper so have not been restated for conciseness.

3. Methods I don't agree with the sentence «qualitative descriptive studies...stay close to the surface of the data». Qualitative research aims to obtain a deeper and more holistic understanding what health professionals, patients or stakeholders think, feel or do in their natural context, through collection and analysis of narrative material, using a flexible study design. Methodology in qualitative research differs in many aspects from quantitative research. In qualitative studies, the questions are broader and open to unexpected discoveries. This flexibility is a strength of qualitative research but only within a coherent methodology as a whole. Qualitative research prefers the question SPIDER (Sample, Phenomenon of Interest, Design, Evaluation, Research Type) over PICO (Population, Intervention, Comparison, Outcome).

Thank you. We understand that this statement is one that could be debated, however the statement is referenced and is therefore not just a statement of opinion. This type of descriptive qualitative study can be classed as *small q* and does not aim for interpretation using a semantic (descriptive) not a latent (interpretative) approach (Braun and Clark 2006, Kidder and Fine 1987). In summary, this qualitative study is part of a larger quantitative intervention study, as such we are aiming to report not to interpret, and in keeping with *small q* research it doesn't align with a specific qualitative paradigm or philosophy.

Analysis: The authors developed an inductive coding scheme, a process of open coding, creating categories and abstraction by formulating a general description of the phenomenon under study: subcategories with similar events and information are grouped together as categories and categories are grouped as main categories. The sampling saturation is reached when no further sampling is

necessary (this should be mentioned by the authors).

Thank you. As you are aware, saturation 'in its true sense' can be a contested term, originally associated with the Grounded Theory approach. We had a convenience sample, and therefore the concept of data saturation was not drawn upon. Instead we drew on the concept of Information Power (Malterud 2016) which takes into consideration the depth of knowledge and understanding provided by information rich participants.

The authors indicated that they used an audit trail as a technique to enhance trustworthiness; they don't say if they used researcher triangulation (all data was analysed involving several authors) which is another technique to enhance trustworthiness.

Researcher (investigator) triangulation was used as multiple researchers were involved in the analysis of the data, please see the discussion provided in lines 172-177 where these concepts are discussed.

4. Results A narrative research approach was followed to describe the results. Table 1 shows that all the participants had their stroke event more than a year ago, at least. This is the first time that the reader is aware of the context of the study: chronic phase after stroke. This is important information that should be in the title, in the methods and in the discussion.

Thank you for highlighting this important point. The reason for not stating the chronic phase in the title has been discussed above. However, we have amended the introduction to clearly state that participants were in the chronic phase (line 131).

5. Discussion The discussion should go into more depth.

Thank you. We have added to the Discussion within the guidance of the journal word count.

The authors don't integrate the results in the theoretic frameworks for Technology Acceptability.

Thank you – as previously stated this study was not designed around a specific framework of acceptability.

The authors don't say anything about the influence of time since stroke. Patients with 1 year since stroke feel the same way than patients with 7 years since their stroke.

Thank you. We agree that time since stroke could be an interesting topic to explore. However, time post stroke was not the focus of this study, and there was no evidence of time post stroke being an influencing factor evident in the qualitative data.

Reviewer: 3

Dr. Simone Dorsch, South Western Sydney Local Health District, Australian Catholic University - North Sydney Campus

Comments to the Author:

Abstract

Clearly written except Results section - particularly lines 64-66 could be re-written to be clearer, maybe specify or provide some example of the Influencing factors, maybe commence sentence with 'Participants' overall experience as influenced by.....' and the following sentence with 'Usability and acceptability of the platform was affected by.....'

Thank you very much for your clear comments and suggestions in your review, they are-appreciated. We agree with your comment on the results section and have reworded it accordingly (see lines 64-66).

Introduction

Clearly written overall. Please include the research aims/questions in further detail

Suggestions are:

Lines 97-98: specify if this is full function of the arm

Noted with thanks and changed "regain full function of the upper-limb at six months"

Lines 105,106 – some terms unclear – what is meant by motor feedback and multisensorial

stimulation

Noted with thanks and changed to be more explicit "...provide immediate feedback on performance, and stimulate the visual, auditory and tactile senses".

Line 107 – should read improvements in performance of daily activities

Noted with thanks and changed as suggested.

Line 109 – larger than what?

Noted with thanks and changed to "...are larger when a gaming component is included compared to VR which only provides feedback without gamification".

Line 120 – contains the terms 'upper-limb' and 'arm' - use consistent terms - I prefer 'arm' because it is more accessible language

Thank you for raising this point. As we have used the term upper-limb in the title we have changed "arm" to "upper-limb" for consistency apart from in direct quotes from participants.

Methods

Clearly written

Comments and suggestions:

I think that the reference 29 describes the method as being 'qualitative descriptive study', rather than the absence of a framework, this statement about not having a methodological framework appears to be contradicted in the Data Analysis section where the use of the Framework method is described

Thank you for highlighting this point. On reflection we can see this as a potential point for confusion and have changed *methodological framework* to *methodological approach*.

Lines 133-134 – more detail about how many participants across what levels of engagement should be provided

Thank you. Information on level of engagement/use is provided in a revised Table 1.

Include ethics clearance details

Thank you, this is provided in line 507 under "ethics approval".

Line 152 – these research aims have not been fully described

Thank you – the research aims are described in lines 129-131 of the clean amended version.

Results

Clearly presented

Suggestions:

Line 286 – improve expression of this sentence

Noted with thanks and changed to "...The most common problem reported with the NeuroBall was damaged, broken or ineffective straps which secure the hand to the device, however resolutions were always found to allow participants to continue with their training."

Line 290 – delete 'as illustrated previously'

Noted with thanks and deleted as suggested.

Line 301 – delete 'as stated'

Noted with thanks and deleted as suggested.

Line 307 – why is it 'perceived monotony' - this is describing a subjective experience and could be written as 'some commenting on monotony'

Noted with thanks and changed to "...despite some reporting that they experienced monotony"

Line 351 – unclear

Noted with thanks, on reflection we felt this sentence did not add to the point being made and have therefore removed it.

Lines 363-364 – is this referring to other research? Needs referencing

Noted with thanks and have provided a reference to Baniña *et al.* 2022 [7] which supports the point

being made.

Lines 382-385 – it would be good to have more detail about these important comments and supporting quotes, if identification of barriers is a research aim then these comments are very important

Noted with thanks. An additional quote has been added to provide more detail and to evidence the point being made “P19 Sam (FMA-UE 35, high user) I don’t think it twists your hand enough (16, 719). I get to the limit and then the machine doesn’t make it any better (17, 724-25).”

Discussion

Clearly written - however some of the concepts discussed here are not evident in the Results

Suggestions:

Line 391 – statement appears contradictory - does it mean that these issues only affected a small number of people or that they did not affect the amount of use?

Noted with thanks and we have reworded this sentence to improve the clarity. It now reads... “While some participants raised issues related to the comfort and ease of use of the device, these mostly did not appear to impact the use of the platform or enjoyment of the games”.

Line 411-412 – needs re-writing to increase clarity

Noted with thanks and changed to “Another factor that appeared as being important in persistent use was the belief that the intervention had the capacity to help, creating psychological investment in the platform (Sekhon 2017)”.

Lines 415-417 – these interesting factors are not well described in the Results section

Noted with thanks, we have added a quote and short description of factors that could have improved motivation within the results in “Theme 4 Factors Motivating Persistence”.

Lines 423-424 – what are these ‘multiple, personal interacting factors’

Noted with thanks, we have provided examples of these factors “Multiple, personal, interacting factors such as preference of game genre, level of difficulty and movements required for the game are likely to contribute to extended gameplay and persistence”.

Line 433 – this is not apparent in the Results section

Noted with thanks, looking at this again we felt it was best to remove this statement in the discussion.

Lines 434-435 – there is a very large body of evidence on neuroplasticity that describes these features of changing function after stroke

Noted with thanks, we acknowledge that whilst there is still much to learn about the drivers of neuroplasticity, for the purposes of this paper there is sufficient literature to support our understanding, therefore we have removed this from the discussion.

Lines 445-446 – unclear

Noted with thanks and changed to “Participants appreciated the clear signposting for support provided by the research team and in the handbook, which is important given other authors have suggested that accessing support may require proactive encouragement [34].

Line 456 – could be the ‘participants were representative of stroke survivors living in the community ...’

Noted with thanks and changed to “Participants were characteristic of many stroke survivors living in the community”

Line 457 – the argument appears to be that the results should be fairly generalisable to similar groups of stroke survivors

Noted with thanks, we have removed “not generalisable”.

Conclusion

Line 474 - typo

Noted with thanks and corrected.

Lines 476-478 – unclear writing

Noted with thanks and have reworded to “Participants highlighted the importance of meaningful motivation to enhance engagement, through features such as challenge, competition, provision of fun

activities, or feedback, which can indicate palpable positive change.”

Lines 478-480 – needs re-writing – what ‘features’ are being described?

Noted with thanks, having reworded the lines in the above comment the ‘features’ being referred to should now be clear.

Acknowledgements - should the stroke survivors involvement with the study design be acknowledged with co-authorship?

Thank you for raising this important point. The stroke survivors on our trial steering group didn't meet and didn't want to meet the requirements of co-authorship. At that time of their involvement, they were happy with being members on the steering committee. We acknowledge however that there has rightly been a change in the expectations of acknowledging participant involvement with co-authorship since this study took place in 2018 and the follow up RHOMBUS II study reflects this.

Reviewer: 4

Dr. Paul Bentley, Imperial College London

Comments to the Author:

Overall comments:

While this work is interesting, I do not feel it is novel, nor does it add new findings or recommendations to the field. The presentation of this work needs to be improved to reach a more appropriate standard for peer-reviewed publication, in terms formatting and grammar, as well as the rigour of scientific reporting standards. In particular, detail is lacking regarding the qualitative methods and analysis.

Thank you, as a fellow developer in this field we are pleased you found our work interesting. We are however not clear on what you base your statement on that you do not find our work novel or that it adds new findings or recommendations to the field. For example, findings from this study add to the field of user experience of specific gaming technologies designed for rehabilitation as opposed to generic/commercially designed devices. This is important given that current literature suggests that specific rehabilitation devices are more effective than “off the shelf” devices.

Likewise, we make specific recommendations for future development of rehabilitation technologies from the feedback received from study participants and for future studies building on this research.

Your comment about the presentation and quality of our written work is out of line with the 3 other reviewers, and you provide no examples of where you think improvements need to be made. Granted there are always some typographical and syntax errors made despite checking, and the other reviewers did highlight a small number of line-by-line suggestions for minor grammatical errors and these have been corrected.

Additional details have been added about qualitative methods and analysis in the collective response to reviewer feedback.

Major comments:

- Please follow CONSORT reporting guidelines (state that this is a feasibility trial) in the title and throughout the manuscript.

Thank you but the CONSORT reporting guidelines were not followed as this study was a) not a randomised trial and b) we have used the Standards for Reporting Qualitative Research (SRQR) guidelines which is in line with the BMJ Open author guidelines (the completed checklist was also submitted as part of the necessary submission documents).

Thank you for your point about feasibility being included in the title. We did consider including the word 'feasibility' however it made the title too long. We think we have made clear that this paper is reporting qualitative findings from a feasibility study. To add further we have edited the abstract and the introduction to stress the feasibility nature of this work.

- The Background does not offer a balanced review and critique of relevant literature, please comment on the quality of evidence presented in systematic reviews and meta-analysis included in your manuscript. It may be worth including some key preclinical evidence and high-quality evidence synthesis (including Cochrane reviews).

Noted with thanks, a more detailed statement has been made on the most recent Cochrane review, however we are restricted by the available word count to provide a detailed discussion of the quality of all reviews cited within this paper.

- Considering this is a quantitative study, there is limited to no discussion of existing qualitative work in the Background. The Background could be improved by exploring the wealth of existing qualitative work in this field and discussing how the present study builds upon this.

We are assuming there was a typographical error in your comment as this is a qualitative study, not quantitative as you state. We are unable to agree with your comment about there being a wealth of existing qualitative work, this is not our experience. We have though added more recent qualitative work in this field to the Background so thank you for raising this point.

- Research aims are not clearly stated within the manuscript. Aims presented within the Abstract are too vague (“To report participants’ experiences of trial processes and use of the Neurofenix platform for home-based rehabilitation following stroke”) and not reflective of the methodology employed.

Thank you for raising the matter about the research aims; this was also commented on by Reviewer 2. As such please refer to responses made to Reviewer 2. We are however unable to agree that the aims are not reflective of the methodology employed as the qualitative methodology is coherent with exploring people’s experiences.

- Methods section- please ensure you follow qualitative reporting standards i.e COREQ checklist. Key information is missing, such as sampling procedures (for both recruitment to the main trial and the purposive sampling method). Please state your theoretical assumptions- ontology, epistemology.

Thank you. As previously stated we have followed the reporting guidelines as directed by the BMJ open guidelines for authors i.e. Standards for Reporting Qualitative Research (SRQR) guidelines, and for which there is a completed checklist. Further details on recruitment for the main feasibility trial are reported in the published paper as referenced. Lines 139 to 143 report details of the purposive sampling procedure (no changes have been made here and remain as stated in the original submission). As previously stated, our study can be considered as *small q* as opposed to *big Q* qualitative research, the former uses the tools of qualitative methods, but without following a specific methodological approach i.e. phenomenology or grounded theory, hence why stating theoretical assumptions would be incongruent with the small q approach taken.

- Methods- no reference to underlying theory used to inform interview guide development or framework development. Please clearly explain what guided the development of these during interactions with PPI representatives- i.e what were some of the key starting blocks.

The topic guide was co-developed by stroke survivors and experts in the field with the goal of being led by participants experiences. An indicative topic guide was created that allowed us to make sure key topics were covered, a copy of which can be seen in the supplementary materials. As can be seen, the topic guide aligned with the study aims with the clear view of wanting to explore participants experiences.

Theoretical models of acceptance have progressed since the time of study design, and as such the

topic guide for the RHOMBUS II trial was developed using the Theoretical Framework of Acceptability (TFA) (Sekhon 2017).

- Analysis- no reference to themes within the analysis, mentions "12 categories", Results section introduces "5 themes". Please discuss how these themes were generated in the analysis section, make clear the distinction between "categories" and "themes".

Thank you. We have taken out “categories” to remove any confusion around categories and themes. In line with standard practice, themes are reported within the results not the analysis.

- Discussion- not in-keeping with study aims or results (limited reference to trial processes). Would benefit from further review of relevant background literature and theory to demonstrate where the current findings sit and what they add.

Thank you. Given the study aimed to explore participant experiences of the trial and using the VR platform we believe the discussion does reflect the study aims and results. Please see the final paragraph of the discussion which covers trial processes, specifically support provision which is the most relevant theme related to trial processes with discussion and reference to relevant background literature.

- Conclusion lacks clarity, and again, is not in-keeping with the main study aims. There is limited critical insight from the authors in terms of limitations of this work. There needs to be more generalisable conclusions/implications drawn for this work to hold relevance to a ~~broad readership~~.

Thank you. Amendments have been made to the conclusion to improve clarity. We feel we have appropriately represented the main findings of the study without over-claiming, and in keeping with the MRC Complex Intervention Framework this is a necessary step in device development.

- Reference to the technology brand (Neurofenix) is featured throughout this manuscript, as well as a link to the company website. I would suggest referring to the brand name in the first instance only, and thereafter, referring to it as a VR platform. Authors competing interests make this particularly pertinent.

Thank you, this is noted, and we have largely removed the brand name as suggested. Please note that academic integrity has not been compromised by any authors competing interests. As someone in this field yourself, you will know you can be a developer and still remain neutral to the reporting of research findings and be named as a co-author.

- Please include coded transcripts, as well as map of codes, sub-themes and themes.

A breakdown of themes and sub-themes is provided in Table 2. In keeping with the nature of qualitative work we did not seek or have consent to publish full transcripts of interviews, this also aligns with the editors concerns around identifiable information about participants personal data which we have addressed in Table 1.

Minor comments:

- Please include a background in the Abstract to provide context for readers

Thank you. We have followed the BMJ open guidance for the format of the abstract which does not ask for background information to be included in the abstract.

- Please label both strengths and limitations and ensure both are equally represented. The first strength listed is a description of the study rather than a strength, please elaborate or clarify this point.

Thank you. BMJ open does not ask for individual labels of strength and limitations, nor does it ask for an equal balance, therefore we have not actioned this comment. We also feel the first strength listed

is a strength given the lack of qualitative data that has explored participation. experiences.

- Please ensure key aims, methods, results and conclusions are succinctly and cohesively presented within the abstract.

Following specific comments from other reviewers the abstract has been strengthened/reshaped which we now feel gives a succinct and coherent account of the study.

- Grammar and presentation need attention to ensure clarity of expression and enhance readability. As other reviewers have kindly identified a small number of minor line by line recommendations, these have been addressed in the resubmitted manuscript.

- Pg4, line 10 “Despite advancements in prevention, treatment and rehabilitation”, please reference We feel this is covered by the provided reference.

- Pg4, line 42, it is not misleading to state that the platform in question is effective in generating improvements in at the level of impairment and function based on a small, non-randomised feasibility study. Please amend.

Thank you. We have not claimed efficacy as this would be incorrect to do so. We have stated that positive effects were demonstrated, which is a *signal* of effect (NIHR do expect signals to be reported). These positive effects are reported in detail in the referenced associated paper.

- Pg5, line 51; Please state PPI guidelines followed, including reimbursement.

Noted, the PPI section has been moved to be above the Data Analysis to hopefully make PPI involvement clearer, and we have now included a reference to the NIHR INVOLVE guidelines which were followed as well as information on reimbursement.

VERSION 2 – REVIEW

REVIEWER	Amorim, Paula University of Beira Interior, Health Sciences Faculty
REVIEW RETURNED	18-Oct-2023

GENERAL COMMENTS	.Title The title doesn't mention the phase after stroke (Chronic phase). 2. Introduction It misses a theoretic basis for Acceptability (e.g., theory of planned behavior, technology acceptance model (TAM), unified theory of acceptance and use of technology [UTAUT]). Additional external factors influence the user acceptance of technology, including demographic characteristics, benefits experienced with technology, technological expertise, and social/cultural influences. Usability, accessibility, comfort, privacy and security, satisfaction, are also important variables to the acceptability of telehealth solutions. The introduction should go into more depth. 3. Methods I don't agree with the sentence «qualitative descriptive studies...stay close to the surface of the data». Qualitative research aims to obtain a deeper and more holistic understanding of what health professionals, patients or stakeholders think, feel or do in their natural context, through
---

	collection and analysis of narrative material, using a flexible study design. Methodology in qualitative research differs in many aspects from quantitative research. In qualitative studies, the questions are broader and open to unexpected discoveries. This flexibility is a strength of qualitative research but only within a coherent methodology as a whole. Qualitative research prefers the question SPIDER (Sample, Phenomenon of Interest, Design, Evaluation, Research Type) over PICO (Population, Intervention, Comparison, Outcome) Main differences between quantitative research and qualitative research Quantitative research Qualitative research Investigation question: PICO Quantitative research asks «how much», «how many», «how often», The investigation question remains the same through the research process Investigation question: SPIDER Qualitative research asks «what», «how», and «why» During the research process the research question might change to a certain degree because data collection and analysis sharpens the researcher's lens. In the methods section you need to describe how and explain why the original research question was changed. Sample and selection: The sample size is larger and calculated with statistical rules in a way to get a representative sample of the population. The ideal is aleatory recruitment. Once the sample size is determined, it is intended to maintain this size until the end of the study; if that is not possible (due to drop-offs, serious secondary effects, etc) in the method section you must explain the reasons. Sample and selection: Small samples and not randomization in the selection process. The choice of participants should be made by individuals or groups who have knowledge on the phenomenon and can articulate and reflect, are motivated to communicate at length and in depth with the investigator and represent the stakeholders. The sample could change during the research process and the size is only closed when information saturation is reached (when it is expected that adding new participants does not add new information but yield redundant information). You should review the sampling plan regularly and adapt when necessary. Comparison between intervention group and control group There is no control group Manipulation of the context occurs Observation in natural context Focus on quantification of predefined variables Open to new findings The aim is generalization of the results to the population Generalization is not intended. The aim is to achieve a deeper understanding of the phenomena There is an effort to avoid researchers' influence (blinding effect) Researchers' influence is inevitable Sampling: the authors have chosen a convenience sampling, selecting the participants that are most easily accessible. Qualitative data collection method: the authors have chosen the interviews as the collection method.
--	---

	Analysis: The authors developed an inductive coding scheme, a process of open coding, creating categories and abstraction by formulating a general description of the phenomenon under study: subcategories with similar events and information are grouped together as categories and categories are grouped as main categories. The sampling saturation is reached when no further sampling is necessary (this should be mentioned by the authors). The authors indicated that they used an audit trail as a technique to enhance trustworthiness; they don't say if they used researcher triangulation (all data was analyzed involving several authors) which is another technique to enhance trustworthiness. 4. Results A narrative research approach was followed to describe the results. Table 1 shows that all the participants had their stroke event more than a year ago, at least. This is the first time that the reader is aware of the context of the study: chronic phase after stroke. This is important information that should be in the title, in the methods and in the discussion. 5. Discussion The discussion should go into more depth. The authors don't integrate the results in the theoretical frameworks for Technology Acceptability. The authors don't say anything about the influence of time since stroke. Patients with 1 year since stroke feel the same way than patients with 7 years since their stroke?
--	---

REVIEWER	Dorsch, Simone South Western Sydney Local Health District, Bankstown-Lidcombe Hospital
REVIEW RETURNED	30-Oct-2023

GENERAL COMMENTS	Congratulations - paper is very clear and easy to read minor suggestions below Abstract Clear No added comments Intro Lines 109-110 – is this one common theme rather than multiple? Lines 114-116 – could be re-worded to be clearer Methods Clear process outlined Results Very clear Lines 339-342 – list too long and unclear expression Discussion Line 406 These findings illustrate that....
--

VERSION 2 – AUTHOR RESPONSE

Reviewer: 2

Dr. Paula Amorim, University of Beira Interior

Comments to the Author:

See the attached file

Thank you for the confirmation that these comments have already been adequately addressed in the previous resubmission.

Reviewer: 3

Dr. Simone Dorsch, South Western Sydney Local Health District, Australian Catholic University - North Sydney Campus

Comments to the Author:

Congratulations - paper is very clear and easy to read

Thank you for your review and your comments.

minor suggestions below

Abstract

Clear

No added comments

Intro

Lines 109-110 – is this one common theme rather than multiple?

Noted with thanks and changed to *“Within qualitative work in the field a common theme that has emerged between studies is the beneficial effect VR has on motivation and engagement with upper-limb rehabilitation”*

Lines 114-116 – could be re-worded to be clearer

Noted with thanks and changed to *“A recent meta-analysis, which compared VR interventions that included gaming components with those which just provided visual feedback, found that the inclusion of gaming components produced larger treatment gains.”*

Methods

Clear process outlined

Results

Very clear

Lines 339-342 – list too long and unclear expression

Noted with thanks and changed to *“Some motivators related to enjoyment, such as playing games connected to previous interests (e.g., football, playing space invaders as a child) and finding gaming more interesting and purposeful than prescribed home exercise programmes. Other motivators were logistical, such as having a structured practice schedule and set amount of time to practice, being able to play at home and having the flexibility to plan practice around daily life.”*

Discussion

Line 406 These findings illustrate that....

Noted with thanks and changed.